# Predicting Chain-of-Thought Correctness from Trajectory Geometry

## Abstract

We ask whether the geometry of a reasoning trajectory, that is, how a chain-of-thought (CoT) trace moves through semantic space beyond its raw length, predicts whether the final answer is correct, and whether that prediction is useful in practice. Across 2,800 CoT traces spanning three reasoning benchmarks (FOLIO, GSM8K, and PrOntoQA) and five language models, we extract interpretable trajectory-level features (adjacent-step transition energy, path entropy, semantic drift, loopiness, discourse-graph spectra, and direction-sensitive drift) and predict per-trace correctness. Under problem-grouped cross-validation, in which every model's traces for a given problem are held out together, the engineered features reach 0.754 balanced accuracy (95% confidence interval [0.735, 0.767]), against 0.589 for a length-only baseline, 0.711 for a mean-pooled sentence-embedding classifier, and 0.677 for a baseline that sees only *which* model produced the trace; the trajectory features thus add about 0.08 balanced accuracy over model identity alone, and the margins over length and embedding are significant under a paired bootstrap ($p < 0.001$) that survives problem-level (clustered) resampling. Our central finding identifies *what* carries the signal: in a within-model ablation that shuffles only the order of a trace's sentences, a sequence model loses 0.081 balanced accuracy, while matched extra capacity over an order-free summary of the same trace adds essentially nothing (about 0.005). The predictive content is therefore specifically the *order* in which a trace moves through semantic space, isolated from trace length and model capacity, and from model identity because the ablation holds the model fixed. Cheap, interpretable trajectory features recover a usable, zero-training-cost fraction of this order signal (a purely geometric subset reaches 0.715, clearly above a lexical baseline at 0.592); a generic order-aware recurrent model recovers more (0.83) at the cost of interpretability and a training step, and the ordering of the engineered features over the baselines holds on a second, architecturally different embedding backbone. Used as a verifier, the classifier reranks the four to five model traces for each problem to a correct answer 88.8% of the time in distribution, against 83.5% for majority voting, though a trivial policy of always trusting the strongest model reaches 0.910; out of distribution the verifier and majority voting fall within a few points of each other on all three held-out benchmarks. The verifier is thus useful in the single-trace, model-unknown setting but does not clearly transfer across benchmarks. We report one further qualifying result, namely that on the synthetic PrOntoQA benchmark a generic embedding baseline matches and slightly exceeds the engineered features.

## 1 Introduction

Chain-of-thought (CoT) prompting (Wei et al., 2022) has made the intermediate reasoning trace a standard artifact of how large language models solve hard problems. A natural question follows: given a trace alone, can we tell whether its final answer is correct? Most existing answers either read the model's internal state or pay for extra samples. We ask a narrower and cheaper question. Does the *geometry* of the trace, the path it follows through semantic space, carry a correctness signal, and is that signal strong enough to act on?

The motivating intuition is simple. A trace that wanders, loops back on itself, or drifts erratically through semantic space may be likelier to be wrong than one that advances steadily toward a conclusion. We operationalize this intuition with a set of interpretable trajectory features computed from sentence embeddings of the trace, and we test, on 2,800 traces from three reasoning benchmarks and five models, whether those features predict per-trace correctness. We then test whether the resulting classifier is useful, by applying it as a post-hoc verifier for selective prediction, for reranking candidate traces, and for early stopping.

Two confounds dominate the design. The first is **length**: longer traces may simply be more (or less) error-prone, so any length-correlated feature inherits that signal trivially. We ablate all length features. The second is **problem leakage**: each problem in our corpus is solved by several models, so a classifier evaluated under ordinary cross-validation can memorize problem identity rather than learn a trace-level signal (Kaufman et al., 2012). We use problem-grouped cross-validation throughout, holding out every model's traces for a problem together.

Our contributions are as follows. (i) We isolate an *order* signal in reasoning trajectories: a within-model ablation that shuffles only sentence order, against a parameter-matched order-free control, attributes the bulk of the predictable signal to order (+0.081 balanced accuracy) rather than to model capacity (+0.005), separating it from length, capacity, and—by construction—model identity. (ii) We show that cheap, interpretable trajectory features recover a usable, zero-training-cost fraction of that signal (a purely geometric subset reaches 0.715, clearly above a lexical baseline), while a generic order-aware recurrent model recovers more (0.83) at the cost of interpretability and a training step—an explicit interpretability/performance trade rather than a claim that the handcrafted features are optimal. (iii) We provide a rigorous methodology for measuring trace-level signal: problem-grouped cross-validation, problem-level (clustered) bootstrap CIs, length residualization, nested model selection, and a second embedding backbone, under all of which the result holds. (iv) We map the boundaries honestly: trajectory features add only about 0.08 over a model-identity baseline; a generic sequence model is the stronger but opaque predictor; and as a verifier the classifier helps in the single-trace, model-unknown regime (0.888 vs 0.835 for majority voting) but is beaten by a trivial strongest-model policy (0.910) and does not clearly transfer out of distribution.

## 2 Related work

Predicting CoT correctness sits among several lines of work. *Verifier and process reward models* train a separate network to score reasoning steps or final answers (Cobbe et al., 2021; Uesato et al., 2022; Lightman et al., 2024). *Self-consistency* samples multiple traces and selects the majority answer (Wang et al., 2023). *Uncertainty and confidence estimation* reads calibrated probabilities or verbalized confidence from the generating model itself (Kadavath et al., 2022; Tian et al., 2023). *Activation probing* predicts correctness or truthfulness from a model's hidden states (Azaria & Mitchell, 2023; Burns et al., 2023). A related line studies whether stated reasoning is faithful to the computation that produced the answer (Turpin et al., 2023; Lanham et al., 2023).

Our setting is deliberately weaker and cheaper than most of these. We use a post-hoc, text-only, model-agnostic classifier over interpretable trajectory features, with no access to logits, hidden states, or extra samples. This paper measures how much correctness signal the bare geometry of a trace contains, under controls that rule out the obvious confounds. We position the engineered features against the natural cheap alternatives, namely length, lexical, and discourse-graph features and a generic sentence-embedding classifier, and we report where an ensemble answer-agreement signal beats all of them.

## 3 Data

Table 1 summarizes the corpus. We use 2,800 CoT traces covering 600 problems drawn from three reasoning benchmarks: FOLIO (Han et al., 2022), a first-order-logic natural-language inference set; GSM8K (Cobbe et al., 2021), grade-school math word problems; and PrOntoQA (Saparov & He, 2023), synthetically generated deduction problems. Each problem is solved by four to five language models (Claude Sonnet, Gemini 2.5 Pro, GPT-OSS-120B, Llama-3.1-8B, and Mistral-7B) at temperature 0, giving one trace per model and problem. Each trace is labeled correct or incorrect by comparing the model's final answer to the benchmark

Table 1: Corpus summary. Each problem is solved by several models, which motivates problem-grouped cross-validation.

| Property | Value |
|---|---|
| Traces | 2,800 (1,845 correct / 955 incorrect) |
| Problems | 600, each solved by 4–5 models |
| Benchmarks | FOLIO, GSM8K, PrOntoQA |
| Models | Claude Sonnet, Gemini 2.5 Pro, GPT-OSS-120B, Llama-3.1-8B, Mistral-7B |
| Decoding | temperature 0 (one trace per model and problem) |

gold answer; for PrOntoQA the gold answer is recomputed independently by forward-chaining the deduction stated in each problem, rather than relying on any precomputed correctness flag in the trace dump.[1]

Each trace is normalized to a common schema, segmented into reasoning sentences, and embedded sentence by sentence with the `all-MiniLM-L6-v2` sentence-transformer model (Reimers & Gurevych, 2019; Wang et al., 2020), which produces a 384-dimensional embedding per sentence. Because each problem recurs across models, every cross-validation split in this paper is grouped by the key `dataset::problem_idx`, so a problem never appears in both the training and the test fold of a split.

## 4    Method

### 4.1    Trace representation

A trace is segmented into reasoning sentences and embedded, giving a sequence $\mathbf{e}_1, \ldots, \mathbf{e}_n \in \mathbb{R}^{384}$. We treat this sequence as a discrete trajectory in semantic space and compute features of its shape. The basic local quantity is the *transition energy* of step $t$, the cosine distance between consecutive sentence embeddings,

$$d_t = 1 - \cos(\mathbf{e}_t, \mathbf{e}_{t+1}), \qquad t = 1, \ldots, n - 1. \tag{1}$$

### 4.2    Feature groups

We compute 52 features in total, organized into named groups. The feature–feature intercorrelations and each feature's correlation with trace length are reported in Appendix F (Figure 3).

- `length_only` (6 features): character, word, and sentence counts, mean sentence length, newline count, and an explicit reasoning-step count.

- `lexical_entropy` (5 features): type-token ratio, repetition rate, unigram and bigram entropy, and the entropy of the sentence-length distribution.

- `marker_features` (17 features): counts of discourse connectives (for example *because*, *therefore*, *wait*, *verify*) and forward and reverse marker-progression indicators.

- `graph_features` (8 features): the trace is turned into a discourse graph whose nodes are sentences and whose edges join sentence pairs with cosine similarity above 0.5; we record density, average degree, clustering, connected components, the Laplacian spectral entropy, and the algebraic connectivity.

---

[1]The trace dump we built on shipped a corrupted PrOntoQA gold field—a constant string for every PrOntoQA trace—so its precomputed correctness flag effectively checked whether the model emitted that string. Recomputing gold by forward-chaining the stated ontology relabels 223 of the 1,000 PrOntoQA traces. We validated the oracle against 15 independent hand and agent re-derivations (15/15 agreement) and against the majority answer of three strong models across all 200 PrOntoQA problems (200/200 agreement); see Appendix D.

- `trajectory_features` (15 features): transition-energy statistics (mean, standard deviation, maximum, minimum), total path energy, direct start-to-end distance, path efficiency, loopiness, semantic-drift slope, path entropy, conclusion compression, two direction-sensitive features and a transition-predictability feature, all described in Section 4.3.

The combined set is `all_features`. We also evaluate the ablations `all_minus_length` (every feature except the length group) and `all_minus_trajectory`, and the partial sets `trajectory_plus_graph` and `trajectory_plus_entropy`. Appendix A gives the full definition of every trajectory feature.

### 4.3 Direction-sensitive features

An earlier version of this work included forward-versus-reverse "energy gap" features. Under symmetric cosine distance these are degenerate. A reversed sentence sequence has the same multiset of adjacent distances, so reverse path energy equals forward path energy and the gap is identically zero. Projection of steps onto the start-to-end axis is likewise invariant under reversal, because reversing the sequence negates both the steps and the axis. We removed these degenerate features and added two that genuinely depend on order. Both are anchored on the *opening* sentence, which is not a symmetric reference point:

- `drift_monotonicity`: the fraction of consecutive steps that move farther from the opening sentence, distinguishing steady outward drift from oscillation.

- `energy_front_back_ratio`: the share of total step displacement that occurs in the first half of the trace, distinguishing front-loaded from back-loaded reasoning effort.

We additionally tested one feature, `path_irreversibility`, that measures whether a trace's step-to-step dynamics are more predictable forward than backward, by fitting affine transition maps in a per-trace principal-component space in both directions and taking the log-ratio of their residual variances. It carries no measurable signal: per-trace estimates from traces of roughly ten steps are too noisy to resolve any asymmetry (Sections 5.2 and 7). We retain it inside `all_features` for continuity with the released artifacts, and Section 5.2 confirms that the headline result does not depend on it.

### 4.4 Classifiers, evaluation, and baselines

We train three classifiers, namely logistic regression, a random forest (Breiman, 2001), and gradient-boosted trees (Chen & Guestrin, 2016), using the scikit-learn implementations (Pedregosa et al., 2011). All evaluation uses 5-fold stratified, problem-grouped cross-validation with median imputation of missing values and class-balanced weighting. We report balanced accuracy and the area under the ROC curve (ROC-AUC). Confidence intervals and paired significance tests use a percentile bootstrap over out-of-fold predictions (Efron & Tibshirani, 1994) with 1,000 resamples; for the paired comparisons we fix a random forest, so that the comparison reflects the feature set rather than an interaction between the feature set and the classifier family.

Baselines come in two kinds. The cheap text baselines are the `length_only`, `lexical_entropy`, and `graph_features` groups above. The key comparison is `embedding_meanpool`: a classifier trained on the 384-dimensional mean-pooled MiniLM sentence embedding of the whole trace. If a generic embedding matches the engineered features, then the feature engineering adds little. We also report `cross_model_answer_agreement`, the fraction of the other models on the same problem that produced the same final answer, which is an ensemble-consistency signal rather than a single-trace one. Two further families of baseline test *what* the predictor uses. To probe whether the signal is order rather than an order-free property of the embedding set, we compare against *order-sensitive* models over the same per-sentence embedding sequence—a small bidirectional GRU and a classifier over the trace's pairwise-cosine matrix (Section 5.2). To probe whether prediction merely reflects which model wrote the trace, we compare against *model-aware* baselines—a one-hot model-identity classifier and a question-text-only classifier (Section 5.6).

Table 2: Correctness prediction, overall, under problem-grouped cross-validation. Balanced accuracy and ROC-AUC are reported for the best of three classifiers per feature group.

| Feature group | Balanced accuracy | ROC-AUC |
|---|---|---|
| majority-class baseline | 0.500 | 0.500 |
| question_only (no trace) | 0.567 | — |
| length_only | 0.589 | 0.659 |
| lexical_entropy | 0.592 | 0.680 |
| graph_features | 0.639 | 0.699 |
| model_identity_only (one-hot model) | 0.677 | — |
| embedding_meanpool (baseline) | 0.711 | 0.780 |
| trajectory_features | 0.716 | 0.817 |
| all_minus_length | 0.752 | 0.867 |
| **all_features** | **0.754** | **0.869** |

Table 3: Balanced accuracy per benchmark, best classifier per cell.

| Benchmark | length | trajectory | all_minus_length | all_features | embedding |
|---|---|---|---|---|---|
| FOLIO | 0.551 | 0.653 | 0.702 | 0.693 | 0.617 |
| GSM8K | 0.605 | 0.722 | 0.738 | 0.750 | 0.624 |
| PrOntoQA | 0.711 | 0.794 | 0.870 | 0.872 | 0.905 |

## 5 Results

### 5.1 Correctness prediction

Table 2 reports overall performance under problem-grouped cross-validation. The engineered features reach 0.754 balanced accuracy, well above the 0.589 of the length-only baseline and the 0.711 of the generic embedding baseline. The `trajectory_features` group alone reaches 0.716, and removing the length features (`all_minus_length`, 0.752) barely changes the headline. For reference, ordinary (non-grouped) cross-validation gives `all_features` 0.743, so the result is not an artifact of problem-identity leakage. Two model-aware baselines put the headline in context (Section 5.6): a classifier that sees only *which* model produced the trace reaches 0.677, and a classifier over the question text alone reaches 0.567, so the trajectory features add roughly 0.08 balanced accuracy over knowing the model and substantially more over knowing only the problem. Part of the headline is therefore a per-model difficulty prior; the order-isolation result of Section 5.2 is a within-model ablation and is *not* subject to this confound, which is why we treat it, rather than the absolute 0.754, as the paper's core finding.

Table 3 breaks the result down by benchmark. The engineered features beat the embedding baseline clearly on FOLIO and GSM8K. On PrOntoQA the embedding baseline is the strongest single method, a point we return to in Section 8.

### 5.2 What the predictive signal is: trajectory order

A predictor over trace features could succeed for several reasons: the signal could lie in trace *length*, in the order-free *content* of the sentence embeddings, or in the *order* in which the trace moves through semantic space. We separate these directly, and find that the predictive content is specifically order. Every comparison in this section uses the same per-sentence MiniLM embedding sequence the trajectory features are computed from, the same problem-grouped folds, and out-of-fold predictions only; the order-aware models are trained per fold with standardisation and early stopping fit on an inner split, never on the test fold.

Table 4: Order-sensitive vs order-free models over the same per-sentence MiniLM sequence (MiniLM, balanced accuracy, problem-grouped CV). The order-aware GRU is the strongest predictor; a parameter-matched order-free MLP gains almost nothing over the linear mean-pool; the weak cosine-matrix model shows it is the recurrent model specifically, not any order representation, that wins; shuffled labels confirm the GRU score is real.

| Model over the trace | Balanced accuracy |
| --- | --- |
| `embedding_meanpool`, linear/forest (order-free) | 0.716 |
| MLP over mean-pool, param-matched (order-free) | 0.722 |
| cosine-matrix $16{\times}16$ (order-derived) | 0.679 |
| `geometry_only` (handcrafted, interpretable) | 0.711 |
| `all_features` (handcrafted, interpretable) | 0.753 |
| GRU over sequence, order shuffled (ablation) | 0.750 |
| **GRU over sequence, order-aware** | **0.831** |
| GRU over sequence, shuffled labels (control) | 0.491 |

Table 5: Within-model decomposition of the gap from an order-free linear baseline to the order-aware GRU. Order is the largest single component; matched capacity over an order-free summary is negligible.

| Step | Balanced accuracy | $\Delta$ |
| --- | --- | --- |
| order-free linear over mean-pool | 0.716 | — |
| + matched capacity (MLP over mean) | 0.722 | +0.005 |
| + set structure (GRU, order shuffled) | 0.750 | +0.028 |
| + **order** (GRU, order-aware) | **0.831** | **+0.081** |

**An order-aware sequence model is the strongest predictor.** A small bidirectional GRU (80,385 parameters) over the per-sentence embedding sequence reaches 0.831 balanced accuracy—above the engineered `all_features` (0.753) and the geometric subset (0.711), and well above the order-free `embedding_meanpool` (0.716) (Table 4). A classifier over the trace's pairwise-cosine matrix, by contrast, reaches only 0.679, so it is the recurrent model specifically, not any order-derived representation, that wins. Permuting the correctness labels collapses the GRU to chance (0.491), so its score is real signal, not leakage.

**The advantage is order, not capacity.** To test whether the GRU simply has more capacity than fourteen handcrafted scalars, we add a parameter-matched, order-free control: an MLP (80,461 parameters) over the same mean-pooled embedding. It reaches 0.722, only 0.005 above the linear mean-pool baseline—extra capacity over an order-free summary buys almost nothing. We isolate the order contribution with a within-model ablation: shuffling the sentence order inside each trace, with the same GRU and the same inputs, drops it from 0.831 to 0.750, a loss of 0.081. Decomposing the gap from the order-free linear baseline to the order-aware GRU (Table 5 and Figure 1), capacity over the mean contributes +0.005, seeing the individual sentence embeddings rather than their mean (still order-free) contributes +0.028, and *order* contributes +0.081, the largest single component. Because this ablation holds the model and the embedding set fixed and changes only order, the order signal is separated from length, from model capacity, and from model identity.

**Interpretable features recover a usable fraction at zero training cost.** The handcrafted trajectory features are not the best predictor, but they are cheap and inspectable. The purely geometric subset (`geometry_only`: the fourteen trajectory features that depend only on the embedding path—transition energy, total path energy, path efficiency, loopiness, semantic drift, conclusion compression, path entropy, and the direction-sensitive features) reaches 0.711 balanced accuracy (0.715 once the inert `path_irreversibility` feature, which carries no measurable signal, is removed), clearly above a lexical-entropy baseline of comparable count (0.592), and recovers most of the order-aware GRU's signal with no

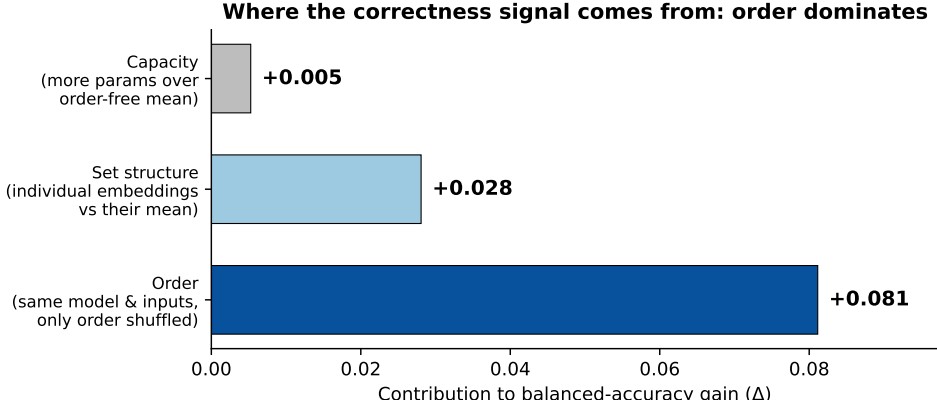

Figure 1: Where the correctness signal comes from. Decomposition of the gain from an order-free linear baseline (mean-pool, 0.716) up to the order-aware GRU (0.831), in balanced accuracy. Extra model capacity over an order-free summary contributes almost nothing (+0.005); seeing the individual sentence embeddings rather than their mean adds a little (+0.028); and *order*—the same model and inputs with only sentence order shuffled—is the dominant component (+0.081). Because the order step holds the model fixed, it is not a model-identity effect.

Table 6: Backbone generalization: balanced accuracy per feature group under three sentence-embedding backbones from different families. `all_features` beats both cheap baselines on all three; `geometry_only` only matches the embedding baseline and flips sign by backbone.

| Feature group | MiniLM-L6 | mpnet-base | e5-base |
|---|---|---|---|
| length_only | 0.589 | 0.589 | 0.589 |
| embedding_meanpool | 0.716 | 0.717 | 0.724 |
| trajectory_features | 0.716 | 0.738 | 0.698 |
| geometry_only | 0.711 | 0.734 | 0.698 |
| **all_features** | **0.753** | **0.765** | **0.743** |

training step and no learned parameters. The trade is explicit: the GRU is more accurate but opaque and must be fit; the trajectory features are weaker but interpretable, training-free, and attributable per feature (Section 7). We therefore do not claim the handcrafted geometry is the best representation—a generic order-aware model captures more—only that it is a cheap, interpretable, effective *partial* recovery of a signal that is fundamentally about order.

### 5.2.1 The ordering holds across embedding backbones

All features derive from sentence embeddings, so we re-extract every feature and re-run the full evaluation under two further encoders from different families, `all-mpnet-base-v2` and `e5-base-v2` (Table 6; every one of the 2,800 traces was embedded by sentence-transformers, verified per trace). The headline ordering is stable: `all_features` beats both `length_only` and `embedding_meanpool` on all three backbones (margin over embedding +0.037, +0.047, +0.019). The geometric subset alone only matches the embedding baseline, and which of the two leads flips by backbone, so we do not claim the geometric features beat a generic embedding; what is backbone-robust is the full engineered set over the cheap baselines, and the order finding itself—the GRU result replicates at 0.833 on `e5-base-v2`.

Table 7: Paired bootstrap on out-of-fold predictions (random forest, 1,000 resamples).

| Comparison | $\Delta$ balanced acc. | 95% CI | $p$ |
|---|---|---|---|
| `trajectory_features` $-$ `length_only` | $+0.128$ | $[0.104, 0.151]$ | $< 0.001$ |
| `all_features` $-$ `length_only` | $+0.165$ | $[0.142, 0.187]$ | $< 0.001$ |
| `trajectory_features` $-$ `embedding_meanpool` | $+0.105$ | $[0.082, 0.127]$ | $< 0.001$ |
| `all_features` $-$ `embedding_meanpool` | $+0.142$ | $[0.121, 0.163]$ | $< 0.001$ |

### 5.3 Statistical significance

Table 7 reports paired bootstrap comparisons on out-of-fold predictions with a fixed random forest. The engineered features beat both the length baseline and the generic embedding baseline by margins whose 95% confidence intervals exclude zero, with $p < 0.001$ in every case. Because each problem is solved by several correlated model traces, we recompute every interval with a *problem-level* (clustered) bootstrap that resamples whole problems rather than individual traces; the intervals barely widen and all six comparisons remain significant, with no flips (Appendix E). We also check that the best-of-three classifier selection is not optimistic: choosing the classifier by nested cross-validation on the training folds only gives `all_features` 0.754, against 0.758 for the reported best-of-three, and the random forest wins all five folds, so the selection does not peek at the test fold.

### 5.4 Robustness

We run four controls; Appendix B gives the full table.

**Shuffled labels (pipeline sanity check).** Permuting the correctness labels collapses performance to chance (`trajectory_features` $0.501 \pm 0.008$, `all_features` $0.497 \pm 0.005$ over repeats). This is a sanity check that rules out leakage and artifactual separability in the pipeline; it is not by itself a proof that every confound is controlled, which is why we test length, model identity, and order separately.

**Length control.** `all_minus_length` (0.752) is statistically indistinguishable from `all_features` (0.754), and both are far above `length_only` (0.589). Going further, residualizing *every* feature on trace length and re-running leaves `all_features` at 0.751 and the geometric subset at 0.716, and the signal persists within each length quartile (0.72 to 0.78 for `all_features`; Appendix F). Length acting through the features does not explain the result.

**Final-answer removal.** Stripping the final-answer span from each trace and recomputing all features, `all_features` holds at 0.696. Even with the final answer removed, the classifier stays close to the generic embedding baseline computed on the full trace (0.711), so it is not merely reading answer formatting.

**Prefix only.** Computed from the first 50% or 75% of each trace, `all_features` scores 0.607 and 0.636 respectively, against 0.738 for the full trace on the same length-filtered subset. Correctness is partially predictable before the trace ends.

### 5.5 Cross-dataset generalization

Table 8 reports leave-one-dataset-out generalization: train on two benchmarks, test on the third. The engineered features beat the length baseline on every held-out split, so the signal is not purely a per-dataset artifact. However, held-out performance (0.58 to 0.79) is below the in-distribution result (0.69 to 0.87 in Table 3). A substantial part of the signal is benchmark-specific, and GSM8K generalizes worst. On held-out GSM8K, adding the lexical, marker, and discourse-graph feature groups hurts generalization (`all_features` 0.579 against 0.640 for `trajectory_features` alone), which suggests that some of the auxiliary features are benchmark-specific.

Table 8: Leave-one-dataset-out balanced accuracy, best classifier per cell.

| Held-out benchmark | length_only | trajectory | all_minus_length | all_features |
|---|---|---|---|---|
| FOLIO | 0.559 | 0.651 | 0.671 | 0.677 |
| GSM8K | 0.528 | 0.640 | 0.597 | 0.579 |
| PrOntoQA | 0.615 | 0.749 | 0.796 | 0.793 |

### 5.6 Model identity and difficulty priors

Because each problem is solved by several models with different overall accuracies, a predictor could in part be reading *which* model wrote the trace rather than the trace's content. We test this directly. A classifier given only a one-hot encoding of the source model reaches 0.677 balanced accuracy, and a classifier over the question text alone reaches 0.567; the trajectory features (0.754) thus add about 0.08 over knowing the model and substantially more over knowing only the problem. A per-model difficulty prior is therefore a real component of the headline, and we report it as such rather than absorb it silently. Two facts keep the central finding intact. First, leave-one-model-out evaluation—train on four models, test on the held-out fifth—averages 0.582 balanced accuracy (Appendix G), so the predictor is substantially model-specific; this is a limitation, not a refutation. Second, and decisively, the order-isolation result of Section 5.2 is a *within-model* ablation: it shuffles sentence order inside each trace with the model held fixed, so the +0.081 order effect cannot be a model-identity artifact. The order signal is the part of the contribution that survives this confound cleanly, which is why we centre the paper on it.

## 6 The verifier use-case

A correctness predictor is only interesting if it can be acted on. We use the classifier (random forest, `all_features`) as a post-hoc verifier for selective prediction, for reranking candidate traces, and for early stopping, and we then test whether the reranking result transfers to an unseen benchmark.

### 6.1 Selective prediction

Ranking traces by classifier confidence and abstaining on the least-confident ones raises accuracy on the accepted set from 0.793 at full coverage to 0.854 at 80% coverage and 0.923 at 50% coverage. The area under the risk-coverage curve is 0.082. Appendix C gives the full curve.

### 6.2 Cross-model reranking, in distribution

For each of the 600 problems we select, among its four to five model traces, the one the verifier scores highest, and we record whether the selected trace has the correct answer. Predictions here are out-of-fold under problem-grouped cross-validation: the verifier never scored a trace whose problem it trained on, but it did train on other problems from the same three benchmarks, so this is an in-distribution measurement. Table 9 compares the policy with random selection, with majority voting over the models' final answers, and with an oracle that succeeds whenever any model produced a correct trace.

In distribution the verifier reranker selects a correct answer 88.8% of the time, 5.3 points above majority voting and most of the way to the oracle ceiling of 95.8%. One baseline bounds this usefulness sharply: a trivial policy that always trusts the single best-overall model reaches 0.910, above the verifier. The verifier's value is therefore specific to the setting it targets—picking among traces when the per-model accuracies are *not* known a priori, for instance a single new model or a pool of unranked models. When a strongest model is identifiable in advance, prefer it; when it is not, the trace-geometry verifier beats the model-blind alternatives (random and majority). Section 6.3 tests whether the in-distribution advantage survives a shift to an unseen benchmark.

Table 9: Cross-model reranking, in distribution: probability of selecting a correct answer among the 4–5 model traces per problem ($n = 600$), with verifier scores out-of-fold under problem-grouped cross-validation. The strongest-model policy always picks the model with the best overall accuracy (a baseline available only when per-model accuracies are known a priori).

| Selection policy | Accuracy |
|---|---|
| random model | 0.655 |
| majority vote (most common final answer) | 0.835 |
| **verifier rerank** | **0.888** |
| strongest-model policy (always the best-overall model) | 0.910 |
| oracle (any correct trace) | 0.958 |

Table 10: Out-of-distribution reranking: the verifier is trained on two benchmarks and reranks the held-out third (200 problems each). The final column repeats the in-distribution reranking accuracy for the same benchmark. Computed by `scripts/10_review_experiments.py`; see the Reproducibility Statement.

| Held-out benchmark | OOD verifier | majority | random | oracle | in-dist. verifier |
|---|---|---|---|---|---|
| FOLIO | 0.745 | 0.740 | 0.624 | 0.950 | 0.805 |
| GSM8K | 0.795 | 0.820 | 0.604 | 0.925 | 0.860 |
| PrOntoQA | 0.980 | 0.945 | 0.738 | 1.000 | 1.000 |

### 6.3 Out-of-distribution reranking

The reranking result above is in-distribution. The deployment scenario that the Broader Impact statement cares about, applying the verifier to a benchmark it was not trained on, is a different regime, and we test it directly: we train the verifier on two benchmarks and rerank the held-out third. Table 10 reports the result.

Out of distribution the in-distribution advantage disappears. The verifier and majority voting fall within roughly three points of each other on every held-out benchmark: the verifier edges FOLIO by 0.5 points, trails majority voting on GSM8K by 2.5 points, and reaches 0.980 against 0.945 on held-out PrOntoQA. The PrOntoQA comparison is near-saturated, however (oracle 1.000, random 0.738, so even random selection scores 0.738), and is therefore uninformative about transfer; the substantive case is GSM8K, where the auxiliary features hurt and majority voting is the better policy. Averaged over the held-out benchmarks the two are within a point of each other, against a 5.3-point verifier lead in distribution. The reranking advantage is therefore an in-distribution effect, and the verifier does not reliably improve over majority voting on a benchmark whose distribution it was not trained on.

### 6.4 Early stopping

We also test committing to a prediction from a trace prefix once a prefix classifier is sufficiently confident. In distribution, at a moderate confidence threshold, the policy reaches 0.783 accuracy against 0.793 for the full trace, while reading 81.1% of each trace on average and stopping early on 42.9% of traces. The compute saving is real but modest.

## 7 Interpretability and a causal probe

**Permutation importance.** Over all 52 features on a held-out grouped split, permutation importance ranks `conclusion_compression` first by a wide margin (a 0.053 drop in balanced accuracy when permuted), followed by `however_count`, `symbolic_token_count`, and `direct_start_end_distance` (Appendix H). The top of the ranking is a mix of trajectory, marker, and lexical features, so no single group carries the result alone; the stratified importances by benchmark and by model are in the appendix.

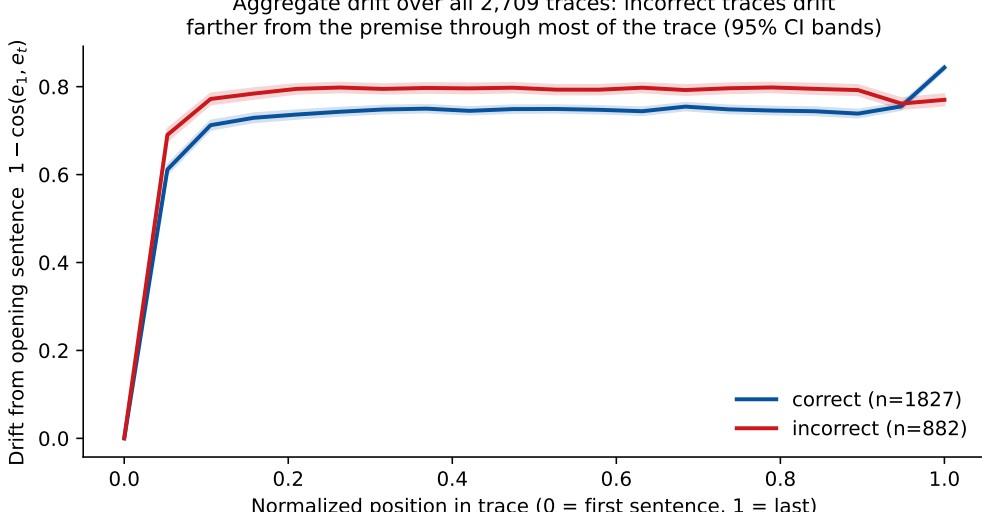

Figure 2: Aggregate (all-trace) drift from the opening sentence, $1 - \cos(\mathbf{e}_1, \mathbf{e}_t)$, by normalized position, split correct vs incorrect over all 2,709 traces with at least three sentences (corrected labels); bands are 95% confidence intervals of the mean. Incorrect traces drift farther from the premise through most of the trajectory. This replaces an anecdotal three-versus-three projection with a population-level view.

**Intervention test.** As a sensitivity analysis, we perturb each trace and recompute its features: shuffling sentence order shifts the predicted correctness probability by 0.32 on average and flips 50% of predictions, and injecting duplicated sentences shifts it by 0.26 and flips 29%, so the classifier responds to structure rather than to an order-invariant summary. This is a sensitivity probe, not a causal claim. The stronger, near-causal evidence that *order* is what matters is the within-model order ablation of Section 5.2, which holds the model and inputs fixed and changes only sentence order.

**Trajectory shape, in aggregate.** Figure 2 shows the population-level version of the intuition that motivates the features: averaged over all 2,709 traces with at least three sentences, the drift from the opening sentence, $1 - \cos(\mathbf{e}_1, \mathbf{e}_t)$, as a function of normalized position in the trace, split by correctness. Incorrect traces sit measurably farther from their opening premise through most of the trajectory—a gap of about 0.05 in cosine distance, with 95% confidence bands that separate from early in the trace onward and stay separated until close to its end—while correct traces stay closer to the premise and then drift to a distinct conclusion at the very end (where, at the final position, they sit farther from the opening than incorrect traces do). This is an aggregate statistic over the whole corpus, not a hand-picked projection; it visualizes the same order-dependent structure that the within-model ablation of Section 5.2 quantifies.

## 8 Negative results

We report two findings that qualify the contribution.

**The embedding baseline wins on PrOntoQA.** On the most regular benchmark, the synthetically generated PrOntoQA, `embedding_meanpool` reaches 0.905 and exceeds `all_features` at 0.872 (Table 3). The advantage of the engineered features is concentrated on the more naturalistic FOLIO and GSM8K, where raw embeddings do markedly worse. The engineering earns its keep on naturalistic reasoning, not on synthetic deduction.

**Ensemble answer-agreement is a strong signal.** As a classifier feature, the fraction of other models on a problem that agree on an answer reaches roughly 0.78 balanced accuracy on its own, above any trajectory feature. It is not a competitor in the single-trace setting that this paper targets, since it requires an ensemble,

and in the reranking task of Section 6 the verifier beats majority voting. Still, any deployment that already has multiple model responses should consider answer-agreement, and a paper on this topic must say so.

## 9 Limitations

**The findings are observational.** The within-model order ablation (Section 5.2) is strong evidence that order, not length or capacity, carries the signal, but it does not identify a mechanism linking a particular kind of trajectory to a particular kind of error. We isolate *that* order matters, not *why*.

**Interpretable features are not the best predictor.** A generic order-aware sequence model outperforms the handcrafted features (0.83 vs 0.75); the trajectory features are a cheap, interpretable, training-free *partial* recovery, not the optimal representation. A practitioner who only wants accuracy should train the sequence model.

**Model-specificity.** Part of the in-distribution headline is a per-model difficulty prior (Section 5.6), and leave-one-model-out transfer is weak (0.582). Its independence from model identity is supported specifically by the within-model order ablation, not by the absolute correctness accuracy.

**Scale and embedding choice.** The study covers three benchmarks, five models, and temperature-0 decoding only; generalization to other task families, larger model sets, and sampled decoding is untested. Features derive from one sentence splitter; the embedding backbone is varied across three encoders (Section 5.2.1) but all are general-purpose sentence encoders.

## 10 Conclusion

The order in which a reasoning trace moves through semantic space carries a real correctness signal, isolated from model identity. We isolate it with a within-model ablation: shuffling only sentence order costs an order-aware model 0.081 balanced accuracy, while matched extra capacity over an order-free summary of the same trace adds essentially nothing, so the predictive content is order rather than length, model capacity, or—because the model is held fixed—model identity. Cheap, interpretable trajectory features recover a usable, zero-training-cost fraction of this signal (0.715 for a purely geometric subset, above the cheap baselines and within reach of the 0.754 full feature set), while a generic order-aware sequence model recovers more (0.83) at the cost of interpretability and a training step. The result is robust: it holds under problem-grouped evaluation, problem-level clustered bootstrap confidence intervals, length residualization, nested model selection, and a second, architecturally different embedding backbone. Used as a verifier the classifier is useful in the single-trace, model-unknown setting—it reranks candidate traces to a correct answer 88.8% of the time, above majority voting and selective-prediction baselines—but a trivial policy of trusting the strongest known model beats it, and it does not clearly transfer across benchmarks. The contribution is therefore a clean isolation of an order signal in reasoning trajectories, a cheap and interpretable partial recovery of it, and an honest map of where it helps and where it does not.

**Broader Impact Statement**

The method is a lightweight, post-hoc filter on reasoning traces. Used well, it can flag low-confidence answers for review and reduce the cost of ensemble inference. The main risk is overtrust: a verifier that is right roughly 89% of the time in the in-distribution reranking setting is still wrong often, and Section 6.3 shows that out of distribution its edge over plain majority voting largely disappears. It should not be used as a sole gate on high-stakes decisions, and it should not be applied to a distribution it was not trained on without first retraining and re-measuring its accuracy on that distribution. The work uses only model-generated reasoning traces on public reasoning benchmarks and involves no human-subjects data.

**Reproducibility Statement**

All numbers in this paper are produced by a single pipeline. The trace corpus, the feature-extraction code,[2] the evaluation scripts, and the verifier and interpretability experiments will be released. Every reported metric corresponds to a named artifact (classifier metrics, ablation tables, robustness reports, cross-dataset metrics, verifier metrics, and interpretability reports) emitted by that pipeline. The order-isolation experiments of Section 5.2, the backbone generalization of Section 5.2.1, the model-aware baselines of Section 5.6, and the clustered-bootstrap, length-residualization, and supplementary analyses (Appendices E, F, G, and H) are produced by a released set of scripts run on the same host and environment as the main pipeline, each writing a named result artifact; they reproduce the committed pipeline figures (for example overall in-distribution reranking 0.888 and `all_features` 0.754) and are consistent with the rest of the paper.

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

## A  Trajectory feature definitions

Let $\mathbf{e}_1, \ldots, \mathbf{e}_n$ be the sentence embeddings of a trace and let $d_t = 1 - \cos(\mathbf{e}_t, \mathbf{e}_{t+1})$ be the transition energy of step $t$. Table 11 defines the 15 features of the `trajectory_features` group.

## B  Full robustness table

Table 12 reports the trace-variant controls of Section 5.4. All rows use problem-grouped cross-validation on the length-filtered subset of 2,714 traces for which every variant is defined, so the `full_trace` row (0.738) is the matched reference for the prefix rows rather than the headline 0.754 of Table 2.

## C  Risk-coverage curve

Table 13 reports the selective-prediction curve of Section 6: accuracy on the accepted set as a function of coverage, when traces are accepted in decreasing order of classifier confidence.

## D  PrOntoQA gold correction

The shared trace dump shipped a corrupted PrOntoQA gold field: the `ground_truth` entry was the constant string `"True"` for every PrOntoQA trace, and the precomputed `correct` flag therefore reduced to checking whether the model's final answer was the literal string `"True"` rather than whether it matched the deduction's gold answer. We replace this with an independent oracle that recomputes gold from each problem's stated ontology by forward chaining (parsing the universal rules, taking the closure of the entity's predicate set, and evaluating the query, including disjunctive, conjunctive, and negated forms). The oracle never reads the dump's `ground_truth` or `correct` fields. Relative to the dump's flag, the corrected labels relabel 223 of the 1,000 PrOntoQA traces (the corrected per-trace correctness rate rises from 0.517 to 0.738). We validated the

Table 11: Definitions of the `trajectory_features` group.

| Feature | Definition |
|---|---|
| `transition_energy_mean/std/max/min` | Mean, standard deviation, maximum, and minimum of the step energies $\{d_t\}$. |
| `total_path_energy` | $\sum_t d_t$, the total cosine distance traversed. |
| `direct_start_end_distance` | $1 - \cos(\mathbf{e}_1, \mathbf{e}_n)$, the straight-line semantic distance from first to last sentence. |
| `path_efficiency` | `direct_start_end_distance` divided by `total_path_energy`; near 1 for a direct path, near 0 for a meandering one. |
| `loopiness` | Mean over sentences $t \geq 3$ of the maximum cosine similarity to any non-adjacent earlier sentence; high when the trace returns near earlier states. |
| `semantic_drift_slope` | Slope of a linear fit of $1 - \cos(\mathbf{e}_1, \mathbf{e}_t)$ against $t$; the rate of outward drift from the opening sentence. |
| `conclusion_compression` | Cosine similarity between the final sentence embedding and the centroid of all preceding sentence embeddings. |
| `path_entropy` | Shannon entropy (base 2) of the normalized step-energy distribution $p_t = d_t / \sum_{t'} d_{t'}$. |
| `drift_monotonicity` | Fraction of consecutive steps for which the distance from the opening sentence increases. |
| `energy_front_back_ratio` | Share of the total step displacement (sum of $\|\mathbf{e}_{t+1} - \mathbf{e}_t\|$) that occurs in the first half of the trace. |
| `path_irreversibility` | Log-ratio of reverse to forward residual variance of an affine transition map fit in a per-trace principal-component space (Section 4.3). Carries no measurable signal; see Section 5.2. |
| `marker_directionality_gap` | Forward minus reverse marker progression: 1 if conclusion markers (*therefore*, *hence*, *finally*, *so*) tend to appear later than premise markers (*because*, *since*, *first*, *second*), and the reverse for the reversed trace. |

Table 12: Balanced accuracy under trace-variant controls, on the length-filtered subset ($n = 2{,}714$) on which every variant is defined. Numbers therefore differ from Table 2.

| Variant | trajectory_features | all_features |
|---|---|---|
| full trace | 0.693 | 0.738 |
| final answer removed | 0.603 | 0.696 |
| first 50% of trace | 0.570 | 0.607 |
| first 75% of trace | 0.585 | 0.636 |

oracle three ways: 15 of 15 hardest items (all false-gold, negation, disjunction, and conjunctive-rule cases) re-derived independently agree with the oracle; the oracle agrees with the majority answer of three strong models (Claude, Gemini, GPT-OSS) on 200 of 200 problems, with the only single-model disagreements being model errors confirmed by hand; and the corrected gold is genuinely mixed (196 True / 4 False at the problem level), not a constant column. PrOntoQA is the only benchmark affected; FOLIO and GSM8K labels were already real correctness grades.

Table 13: Selective accuracy at selected coverage levels (`all_features`, random forest).

| Coverage | 0.10 | 0.25 | 0.50 | 0.65 | 0.80 | 0.90 | 1.00 |
|---|---|---|---|---|---|---|---|
| Selective accuracy | 0.996 | 0.983 | 0.923 | 0.895 | 0.854 | 0.826 | 0.793 |

# E    Clustered bootstrap and nesting audit

The shipped bootstrap resamples at the trace level. Because each problem is solved by several correlated model traces, we recompute every confidence interval with a problem-level (clustered) bootstrap that resamples whole problems (cluster = `dataset::problem_idx`). Table 14 shows the paired significance comparisons: the clustered intervals barely widen and all six remain significant, with no flips. Tables 15 and 16 give the per-benchmark, leave-one-dataset-out, and reranking intervals under both schemes. The best-of-three classifier selection is not optimistic: choosing the classifier by nested cross-validation on the training folds only gives `all_features` 0.754 against 0.758 for the reported best-of-three, and the random forest wins all five folds.

Table 14: Paired bootstrap, trace-level vs problem-level (clustered) resampling, $\Delta$ balanced accuracy. No comparison loses significance under clustering.

| Comparison | $\Delta$ | trace 95% CI | clustered 95% CI | sig? |
|---|---|---|---|---|
| trajectory − length | +0.128 | $[0.104, 0.151]$ | $[0.105, 0.153]$ | yes |
| all_features − length | +0.165 | $[0.142, 0.187]$ | $[0.142, 0.190]$ | yes |
| all_minus_length − length | +0.163 | $[0.141, 0.185]$ | $[0.139, 0.187]$ | yes |
| all_minus_length − embedding | +0.140 | $[0.118, 0.161]$ | $[0.116, 0.162]$ | yes |
| trajectory − embedding | +0.105 | $[0.082, 0.127]$ | $[0.082, 0.129]$ | yes |
| all_features − embedding | +0.142 | $[0.121, 0.163]$ | $[0.118, 0.164]$ | yes |

Table 15: Per-benchmark and leave-one-dataset-out `all_features` balanced accuracy with trace-level and clustered 95% CIs.

| Setting | Benchmark | Point | trace CI | clustered CI |
|---|---|---|---|---|
| per-benchmark | FOLIO | 0.695 | $[0.667, 0.724]$ | $[0.665, 0.726]$ |
| per-benchmark | GSM8K | 0.703 | $[0.671, 0.736]$ | $[0.667, 0.739]$ |
| per-benchmark | PrOntoQA | 0.865 | $[0.839, 0.893]$ | $[0.837, 0.890]$ |
| leave-one-out | FOLIO | 0.677 | $[0.650, 0.707]$ | $[0.645, 0.708]$ |
| leave-one-out | GSM8K | 0.546 | $[0.522, 0.570]$ | $[0.522, 0.567]$ |
| leave-one-out | PrOntoQA | 0.793 | $[0.763, 0.823]$ | $[0.762, 0.825]$ |

# F    Length residualization

To test whether the signal is trace length acting through the geometry features, we residualize every feature on `trace_char_len` (replace each feature with the residual of a linear regression on length) and re-run the grouped-CV evaluation, and separately evaluate within length quartiles (Table 17). Residualizing length out barely moves either group (`all_features` 0.754 → 0.751, `geometry_only` 0.711 → 0.716), and the signal persists within every quartile. Several path features do correlate with length (`total_path_energy` $r$=0.76, `path_entropy` $r$=0.72), but removing that correlation does not remove the predictive signal.

Table 16: Reranking accuracy with problem-level 95% CIs (the natural unit for reranking is the problem). In distribution the verifier's interval clears majority voting's; out of distribution the two overlap on every benchmark.

| Setting | verifier (CI) | majority (CI) |
|---|---|---|
| in-distribution | 0.888 [0.863, 0.913] | 0.835 [0.807, 0.863] |
| OOD held-out FOLIO | 0.745 [0.685, 0.805] | 0.740 [0.675, 0.795] |
| OOD held-out GSM8K | 0.795 [0.740, 0.845] | 0.820 [0.765, 0.870] |
| OOD held-out PrOntoQA | 0.980 [0.960, 0.995] | 0.945 [0.915, 0.975] |

Table 17: Length residualization and within-quartile evaluation, balanced accuracy.

| Condition | all_features | geometry_only |
|---|---|---|
| baseline | 0.754 | 0.711 |
| residualized on length | 0.751 | 0.716 |
| quartile Q1 (char-len 3–638) | 0.778 | 0.751 |
| quartile Q2 (640–965) | 0.721 | 0.679 |
| quartile Q3 (966–1417) | 0.763 | 0.690 |
| quartile Q4 (1419–5417) | 0.750 | 0.709 |

## G  Model-aware baselines

Table 18 gives the model-aware prediction and reranking baselines of Section 5.6, and the leave-one-model-out correctness transfer. Model-identity-only reaches 0.677 and question-only 0.567 for prediction; leave-one-model-out correctness averages 0.582, with the lowest values on the most class-imbalanced held-out models (Gemini 0.517, Mistral 0.513, both near a single dominant class). For reranking, a trivial strongest-model policy (0.910) exceeds the verifier (0.888).

Table 18: Model-aware baselines (problem-grouped CV). Left: prediction balanced accuracy. Right: leave-one-model-out correctness (train on four models, test on the held-out fifth).

| Prediction baseline | bal. acc | Held-out model | bal. acc |
|---|---|---|---|
| all_features (reference) | 0.754 | Claude Sonnet | 0.657 |
| model-identity-only | 0.677 | Gemini 2.5 Pro | 0.517 |
| question-only (no trace) | 0.567 | GPT-OSS-120B | 0.645 |
| | | Llama-3.1-8B | 0.576 |
| | | Mistral-7B | 0.513 |
| | | **mean** | **0.582** |

## H  Supplementary analyses

**All three classifiers per feature group.** Table 19 reports every classifier family for every feature group, rather than the best-of-three of Table 2. The random forest is generally strongest, and the ordering of feature groups is stable across classifiers.

**Permutation importance.** Over all 52 features on a 30% grouped held-out split ($n_{\text{repeats}}$=10), the largest balanced-accuracy drops are conclusion_compression (+0.053), however_count (+0.036), symbolic_token_count (+0.016), and direct_start_end_distance (+0.013); the full ranking and the

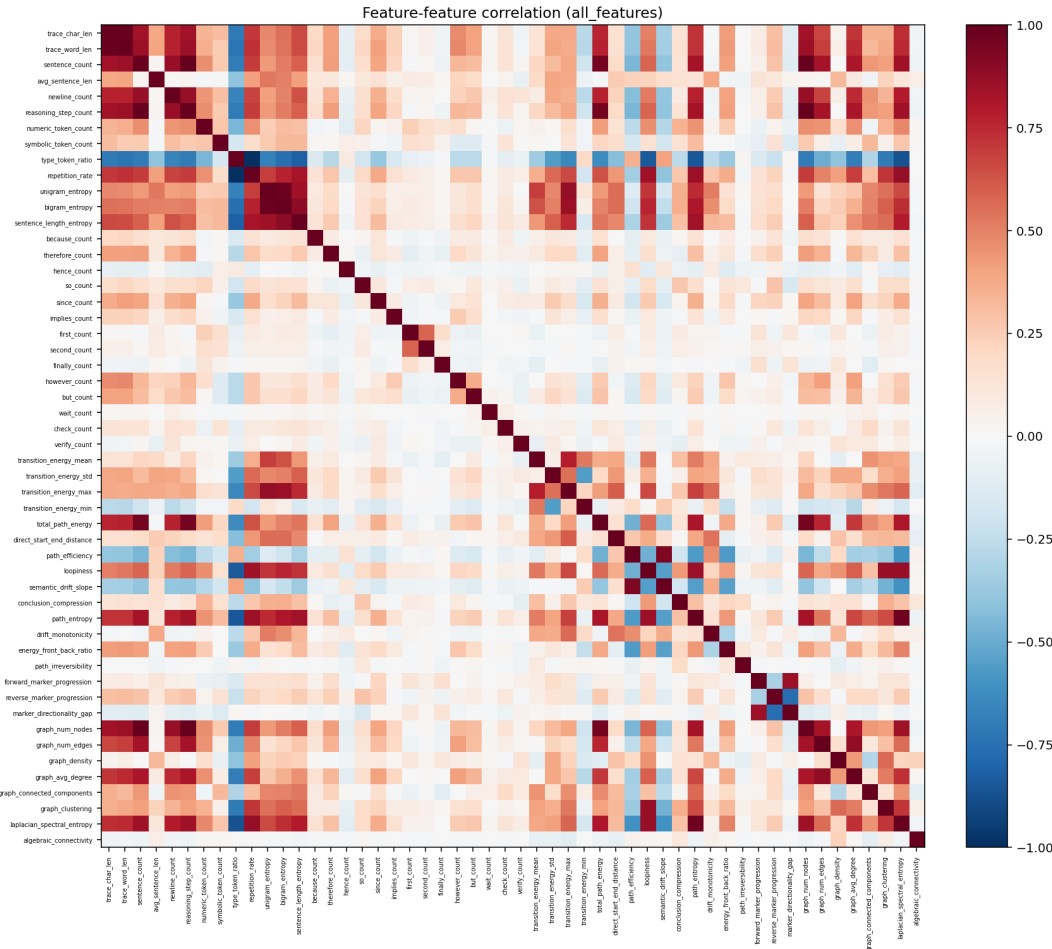

Figure 3: Feature–feature Pearson correlation matrix over the `all_features` group. The length features (`trace_char_len`, `trace_word_len`, `sentence_count`) and the length-correlated path features (`total_path_energy` $r=0.76$, `path_entropy` $r=0.72$ with length) are visible as a correlated block; per-feature correlations with trace length are released as a result artifact. Residualizing length out leaves the predictive signal essentially unchanged (Table 17), so these intercorrelations with length do not drive the result.

by-benchmark and by-model stratifications are released as result artifacts. The top of the ranking mixes trajectory, marker, and lexical features, so no single feature group carries the result alone.

Table 19: Balanced accuracy / ROC-AUC for all three classifier families per feature group (overall, problem-grouped CV).

| Feature group | logistic reg. | random forest | XGBoost |
|---|---|---|---|
| length_only | 0.583/0.616 | 0.589/0.659 | 0.573/0.688 |
| lexical_entropy | 0.522/0.547 | 0.591/0.644 | 0.592/0.680 |
| marker_features | 0.656/0.731 | 0.694/0.755 | 0.613/0.759 |
| graph_features | 0.639/0.699 | 0.615/0.672 | 0.592/0.705 |
| trajectory_features | 0.680/0.747 | 0.716/0.817 | 0.706/0.799 |
| all_minus_length | 0.743/0.819 | 0.752/0.867 | 0.738/0.856 |
| all_features | 0.739/0.819 | 0.754/0.869 | 0.741/0.860 |
| embedding_meanpool | 0.711/0.780 | 0.611/0.776 | 0.651/0.783 |
| answer_agreement | 0.781/0.822 | 0.781/0.813 | 0.767/0.811 |

