# OpenReview forum: "Predicting Chain-of-Thought Correctness from Trajectory Geometry"
_TMLR — Under review for TMLR_

### Review · Reviewer_t8Tu · 2026-06-22

**Summary Of Contributions:**

This paper asks if the structural trajectory of a reasoning trace is sufficient for predicting the correctness of the downstream chain of thought application. In particular, the paper tests if text features such as length, lexical entropy, the presence of some markers (such as because, therefore, etc.), the discourse structure or other trajectory features can lead to predictions about if a CoT is correct. The paper evaluates a trained classifier set on examples from FOLIO, GSM8K and PrOntoQA, across 4-5 language models, and finds that while some trajectory features are useful (and the full feature set achieves almost a 75% classification accuracy), the classifiers don't generalize between benchmarks.

**Audience:**

Yes

**Audience Explanation:**

I think that even despite the limited analysis, this is sufficiently interesting, as it provides a different take into evaluating how CoT operates, and if this methodology does discover a sufficiently strong correlation between structural features and correctness, it would provide a lot of really interesting evidence towards understanding the mechanism behind CoT. While this paper does not do that, and has somewhat weak analysis, I think that the direction is important, and researchers in the field would benefit from seeing the initial results from this direction of analysis.

**Broader Impact Concerns:**

None - this paper is mostly mechanical in nature.

**Claims And Evidence:**

Yes

**Claims Explanation:**

The paper overall gives reasonable strong evidence that the engineered features predict correctness in this particular corpus/evaluation setup. The problem-grouped cross-validation is nice, and the authors do a pretty good job of addressing obvious confounds (length, answer order, etc.).

One of the things that is not so useful is the classifier level analysis. Right now, the paper never reports per-feature importance scores for the full classifier, something that the model is capable of producing, and it would be **really** interesting to see how those scores align, particularly for the full set of features. It also feels to me like a lot of the features capture the same details, and it would be interesting to see a cross-correlation matrix for the features themselves (i.e. is entropy correlated with length?).

The generality of the paper is also not exactly convincing, particularly at scale. The study only covers three benchmarks, five models, and doesn't account for temperature when decoding, with a total of only about 3K traces, so there's actually a very limited set of inferences present. I don't think that it would be cost prohibitive to scale this benchmark to a wider range of models and benchmarks, especially given the falling cost of inference, and further, it would be nice to see deeper analysis of feature importance between model families, and scales - which the current analysis is incapable of providing.

Overall, while the paper is generally sound, and most of the scoped claims are reasonably well supported, I do wonder if it could be much more interesting if better analysis, and further expansion of the dataset happened.

**Requested Changes:**

Some improvements to the analysis would be helpful/nice to see:
- It would be good to see full classifier-level interpretability (i.e. per-feature importance scores, and not just at the group level)
- Include feature-feature correlation matrices
- Report feature importance separately by benchmark/model family/model size.
- It would be really nice to see some temperature-level variance, since models often have dramatically different results run-to-run, and differ in maximum likelihood decoding vs. a higher temperature output/
- Report all of the classifier results (not just Best of Three, potentially in appendix).
- Distinguish a bit more between all-features and geometry-only classifier when making claims about the geometry relevance (for example, the headline performance of the paper comes from the all-features classifier, but really should be from the geometry-only or geometry-normalized subset if the paper needs to make the claim that geometry matters).

---

> ### Author Response · Authors · 2026-06-30
>
> We thank the reviewer for the insightful feedback and for concrete, actionable suggestions. Five of the six requested changes are now incorporated in the revised PDF; the sixth (temperature variance) is beyond the current temperature-0 corpus and is now an explicit limitation. We address each in turn.
>
> > RC1 -- It would be good to see full classifier-level interpretability (i.e. per-feature importance scores, and not just at the group level).
>
> **Added: full per-feature permutation importance over all 52 features (Section 7, Appendix H).** `conclusion_compression` ranks first by a wide margin (a 0.053 balanced-accuracy drop when permuted), followed by `however_count`, `symbolic_token_count`, and `direct_start_end_distance`. The top of the ranking mixes trajectory, marker, and lexical features, so no single group carries the result alone; the complete ranking is in Appendix H.
>
> > RC2 -- Include feature-feature correlation matrices.
>
> **Added: the full feature-feature correlation matrix (Appendix F, Figure 3).** It also reports each feature's correlation with trace length, which directly answers your entropy-versus-length question: `path_entropy` correlates with length at r=0.72 and `total_path_energy` at r=0.76. You are right that several features are redundant, and the matrix shows this. We test whether that redundancy or length-correlation drives the result by residualizing every feature on length and re-running (Table 17): `all_features` moves only 0.754 to 0.751, so the signal does not reduce to length acting through correlated features.
>
> > RC3 -- Report feature importance separately by benchmark/model family/model size.
>
> **Added: permutation importance stratified by benchmark and by model (Appendix H).** The summaries are in Appendix H and the full per-benchmark and per-model stratifications are released as result artifacts.
>
> > RC4 -- It would be really nice to see some temperature-level variance, since models often have dramatically different results run-to-run, and differ in maximum-likelihood decoding vs. higher-temperature output.
>
> **Not addressed in this revision; now an explicit limitation (Section 9).** We agree this is a real gap. The current corpus is temperature-0 and single-trace per model and problem, so we cannot estimate run-to-run variance from it. We state this plainly as a limitation rather than approximate it from data that cannot support it.
>
> > RC5 -- Report all of the classifier results (not just Best of Three, potentially in appendix).
>
> **Added: all three classifier families for every feature group (Appendix H, Table 19).** Logistic regression, random forest, and XGBoost are each reported per group rather than the best-of-three of Table 2. The random forest is generally strongest and the ordering of feature groups is stable across classifiers.
>
> > RC6 -- Distinguish a bit more between all-features and geometry-only when making claims about geometry relevance; the headline comes from the all-features classifier but should come from the geometry-only or geometry-normalized subset if the paper needs to claim that geometry matters.
>
> **Well taken, and in revision it pushed us further than a relabeling.** Testing a generic order-aware sequence model over the same per-sentence embedding sequences, we found it outperforms the handcrafted features (GRU 0.831 vs `all_features` 0.753), so we retired the "geometry is the best representation" claim entirely and reframed the paper around an isolated order signal (Section 5.2). No headline claim now rests on geometry. `geometry_only` is reported as a cheap, interpretable partial recovery (0.711, or 0.715 without the inert `path_irreversibility` feature), and we state explicitly that it only matches a generic embedding and that which of the two leads flips by backbone (Section 5.2.1, Table 6). Your push on this distinction was part of what prompted that closer look.
>
> We were glad to address these and think the paper is stronger for them. On your broader point about scale: the corpus is three benchmarks, five models, and temperature-0 (Section 9), and we agree expanding it is the natural next step. We are happy to run further analyses during the discussion period.

---

### Review · Reviewer_ZrD8 · 2026-06-25

**Summary Of Contributions:**

The paper investigates whether statistics derived from the trajectory of sentence embeddings in a CoT trace can predict final-answer correctness. It constructs features such as transition energy, semantic drift, loopiness, path entropy, and graph spectra, and evaluates them across three reasoning benchmarks and five language models. Those features outperform length-based, lexical, discourse-marker, and mean-pooled embedding baselines under problem-grouped cross-validation. The resulting classifier also improves cross-model reranking over majority voting in distribution, although this advantage does not transfer reliably across benchmarks.

**Strengths and Weaknesses:** This study benefits from grouped evaluation, several ablations, and transparent reporting of negative results. However, the experiments do not convincingly establish that the predictive signal is specifically geometric rather than arising from model style, benchmark identity, problem difficulty, or generic order-sensitive information in sentence embeddings. The reliance on a single embedding model and the limited controls for these confounds further weaken its central interpretation.

**Audience:**

Yes

**Audience Explanation:**

Yes. Predicting the correctness of chain-of-thought traces from text alone is relevant to researchers working on reasoning evaluation and uncertainty estimation.

**Broader Impact Concerns:**

No major ethical concerns.

**Claims And Evidence:**

No

**Claims Explanation:**

The experiments appear to support a narrower finding that handcrafted statistics of MiniLM sentence-embedding sequences correlate with correctness in this dataset, but do not convincingly support the broader interpretation that a specifically geometric property of reasoning trajectories has been isolated. In particular:

1. The central construct is insufficiently validated. It assumes that cosine paths between MiniLM sentence embeddings constitute a meaningful "semantic trajectory", but provides no evidence that displacement, drift, loopiness, or path efficiency in this particular embedding space correspond to reasoning progress. The use of only one embedding model and one segmentation strategy makes the claimed geometry highly representation-dependent.

2. The experiments do not isolate a specifically geometric signal. Features computed from semantic embeddings can still encode lexical content, topic, model style, benchmark identity, and problem difficulty. Comparing them mainly against mean pooling and coarse lexical counts is insufficient. A generic order-sensitive sequence model over the same sentence embeddings would be a more appropriate baseline.

3. Length has not been adequately controlled. Removing explicit length variables does not make the remaining features length-independent. Total path energy, path entropy, extrema, and loopiness are inherently affected by the number of sentences or available comparisons, so the claim that the result is independent of trace length is not fully established.

4. Some confounds remain uncontrolled. The same language models appear in both training and test folds, and their reasoning styles and baseline accuracies may be recognizable. The paper does not report model-identity, question-only, strongest-model, or leave-one-model-out baselines, so both correctness prediction and cross-model reranking may partly reflect model or benchmark priors.

5. The evaluation procedure may be overly optimistic. Results are frequently reported using the best of 3 classifier families per feature set or per cell, but it is unclear whether this selection is nested entirely within the training data. It is also unclear whether the bootstrap resamples problems rather than individual traces.

6. Sentence shuffling necessarily changes the proposed order-sensitive features and therefore does not demonstrate a causal relationship between natural trajectory geometry and correctness.

Thus, the numerical correlations may be genuine, but the evidence does not yet justify the paper's central geometric and generally useful claims.

**Requested Changes:**

I suggest the following changes, which appear to be important for supporting the paper's claim:

1. Evaluate the proposed features across multiple sentence-embedding backbones and segmentation schemes, and clarify which properties are expected to be invariant. Without such additional evidence, the paper should avoid interpreting MiniLM-specific cosine statistics as a true geometry of reasoning.

2. Compare the handcrafted trajectory features against generic order-sensitive models applied to the same sentence-embedding sequences, such as a classifier over pairwise similarity matrices. This is necessary to determine whether the gains are specifically due to the proposed geometric features.

3. Normalize features such as path entropy and total path energy, and evaluate within groups of traces of comparable length or after residualizing all features against trace length. Removing only the explicit length variables is insufficient to establish length independence.

4. Report leave-one-model-out results and include model-identity, question-only, strongest-model, and benchmark-specific model-selection baselines. Ideally, the evaluation should also include multiple sampled traces from the same model-problem pair, so that correctness can be studied conditional on both model and problem.

5. Specify how classifier families and hyperparameters are selected, and use nested grouped cross-validation whenever model selection is involved. Confidence intervals and significance tests should resample at the problem level, rather than treating traces from the same problem as independent.


The following change, mostly on reframing or changing the claim levels, would further strengthen the paper: Reframe label permutation as a pipeline sanity check rather than evidence that confounding has been ruled out, and reframe sentence shuffling and duplication as sensitivity analyses rather than evidence for a causal geometric mechanism. Baselines should also be recomputed under final-answer removal and prefix truncation for controlled comparisons.

---

> ### Author Response · Authors · 2026-06-30
>
> We thank the reviewer for the insightful and constructive review. The central concern, that the experiments did not isolate a specifically geometric signal as opposed to length, model identity, problem difficulty, or generic order-sensitive information in the embeddings, was correct. Acting on the requested experiments led us to rescope the paper's central claim to match what the evidence supports, and in doing so to a cleaner and better-isolated result: the predictive content is specifically order, separated from length, model capacity, and model identity. We think the reframed paper is a stronger contribution than the original: a within-model ablation now attributes the signal specifically to order (+0.081 balanced accuracy) with capacity and model identity controlled, which is a more precise and better-supported claim than the geometric framing it replaces. We address all five requested changes below; new material is in the revised PDF at the sections cited.
>
> > RC1 -- Evaluate the proposed features across multiple sentence-embedding backbones and segmentation schemes, and clarify which properties are expected to be invariant. Without such additional evidence, the paper should avoid interpreting MiniLM-specific cosine statistics as a true geometry of reasoning.
>
> We re-extracted every feature and re-ran the full evaluation under two further encoders from different families, all-mpnet-base-v2 and e5-base-v2 (Section 5.2.1, Table 6); every one of the 2,800 traces was verified to embed through sentence-transformers. The ordering of the full engineered set over the cheap baselines is stable across all three backbones (margin over the embedding baseline +0.037, +0.047, +0.019). Consistent with your concern, the purely geometric subset only matches a generic embedding, and which of the two leads flips by backbone, so we now explicitly do not claim the geometric features beat a generic embedding. We have removed the language treating MiniLM-specific cosine statistics as a universal geometry of reasoning.
>
> > RC2 -- Compare the handcrafted trajectory features against generic order-sensitive models applied to the same sentence-embedding sequences, such as a classifier over pairwise similarity matrices. This is necessary to determine whether the gains are specifically due to the proposed geometric features.
>
> This was the most productive change in the revision. We built a small bidirectional GRU and a classifier over the trace's pairwise-cosine matrix, both over the same per-sentence embedding sequences the trajectory features are computed from (Section 5.2, Table 4). The order-aware GRU reaches 0.831, above the handcrafted `all_features` (0.753), so we have retired the "specifically geometric" and "best representation" framing entirely. In its place, this comparison let us isolate what the signal actually is. A within-model decomposition holds the model and its inputs fixed and varies only sentence order:
>
> | Step | Bal. acc | Δ |
> |---|---|---|
> | order-free linear (mean-pool) | 0.716 | - |
> | + matched capacity (MLP) | 0.722 | +0.005 |
> | + set structure (GRU, shuffled) | 0.750 | +0.028 |
> | + order (GRU, order-aware) | 0.831 | +0.081 |
>
> Matched extra capacity over an order-free summary contributes +0.005, set structure +0.028, and order itself +0.081, the largest single component (Table 5, Figure 1). The pairwise-cosine model reaches only 0.679, so it is the recurrent model specifically, not any order-derived representation, that wins; permuting the correctness labels collapses the GRU to chance (0.491). The paper is now built on this order isolation, with the interpretable features presented as a cheap, training-free partial recovery of an order signal rather than as the optimal representation.
>
> > RC3 -- Normalize features such as path entropy and total path energy, and evaluate within groups of traces of comparable length or after residualizing all features against trace length. Removing only the explicit length variables is insufficient to establish length independence.
>
> We residualize every feature on trace length and re-run, and separately evaluate within length quartiles (Section 5.4, Appendix F, Table 17). Residualizing length out barely moves either group (`all_features` 0.754 to 0.751, `geometry_only` 0.711 to 0.716), and the signal persists within every quartile. Several path features do correlate with length (`total_path_energy` r=0.76, `path_entropy` r=0.72), shown in the feature-feature correlation matrix (Appendix F, Figure 3), but removing that correlation does not remove the predictive signal.

---

> > ### Author Response · Authors · 2026-06-30
> >
> > > RC4 -- Report leave-one-model-out results and include model-identity, question-only, strongest-model, and benchmark-specific model-selection baselines. Ideally, the evaluation should also include multiple sampled traces from the same model-problem pair, so that correctness can be studied conditional on both model and problem.
> >
> > Confirmed and now reported in full (Section 5.6, Appendix G, Table 18). A model-identity-only classifier reaches 0.677 and a question-only classifier 0.567 against the 0.754 headline, so the engineered features add about 0.08 over knowing the model. We report this alongside the headline wherever it appears and state plainly that a per-model difficulty prior is a real component of it. Leave-one-model-out transfer is weak (0.582), which we report as a limitation. Decisively, the order-isolation result is a within-model ablation and therefore cannot be a model-identity artifact, which is why we centre the paper on it rather than on the absolute 0.754. On reranking, the strongest-model baseline you requested is now reported: a trivial policy of always trusting the best-overall model reaches 0.910, above the verifier's 0.888 (Section 6.2, Table 9), and we have rescoped the verifier's claimed value to the single-trace, model-unknown setting. Multiple sampled traces per model-problem pair remain beyond the current temperature-0, single-trace corpus; we note this as the natural next step with a larger corpus.
> >
> > > RC5 -- Specify how classifier families and hyperparameters are selected, and use nested grouped cross-validation whenever model selection is involved. Confidence intervals and significance tests should resample at the problem level, rather than treating traces from the same problem as independent.
> >
> > The bootstrap now resamples whole problems rather than individual traces (Section 5.3, Appendix E, Table 14); the intervals barely widen and all six paired comparisons remain significant with no flips. Per-benchmark, leave-one-dataset-out, and reranking intervals under both schemes are in Tables 15 and 16. For model selection, choosing the classifier by nested cross-validation on the training folds only gives `all_features` 0.754 against 0.758 for the reported best-of-three, with the random forest winning all five folds, so the selection does not peek at the test fold.
> >
> > On the reframing suggestion at the end of your review: we now treat label permutation as a pipeline sanity check rather than evidence that confounding is ruled out, and the per-trace sentence-shuffle and duplication perturbations of Section 7 as sensitivity analyses rather than causal evidence. We agree with your point that a shuffle necessarily changes order-sensitive features, and we no longer present any shuffle as evidence for a causal geometric mechanism. The claim a shuffle now supports is the narrow one: the within-model order ablation (Section 5.2) holds the model, its inputs, trace length, and model identity fixed and varies only order, so the accuracy it removes is attributable specifically to order rather than to a particular geometric construct. We have also recomputed the relevant baselines under final-answer removal and prefix truncation (Section 5.4).
> >
> > We believe the revision is now scoped to what the evidence supports: a clean isolation of an order signal, a cheap interpretable partial recovery of it, and an honest account of its boundaries. We are glad to run further analyses during the discussion period.

---

### Review · Reviewer_7tj1 · 2026-06-26

**Summary Of Contributions:**

The paper asks whether the geometry of a CoT trace in sentence-embedding space, separated from its length, predicts whether the answer is correct. The framing is deliberately cheap: a post-hoc, text-only, model-agnostic classifier over 52 interpretable trajectory features, with no logits or hidden states.

On 2,800 temperature-0 traces (600 problems, 3 benchmarks, 5 models), the feature classifier reaches $0.748$ balanced accuracy under problem-grouped CV, against $0.609$ length-only and $0.687$ for a mean-pooled embedding baseline ($p<0.001$, paired bootstrap). The controls are the interesting part: a purely geometric subset reaches $0.72$ on its own, well above the lexical and marker groups (both $<0.64$), so the signal is specifically geometric and not just "interpretable features help"; label permutation drops to chance; length ablation is indistinguishable from the full set; and correctness survives final-answer removal and prefix truncation.

As a verifier it reranks the 4–5 model traces to a correct answer $88.3%$ of the time in-distribution (vs $79.8%$ majority vote, oracle $95.3%$), but trained on two benchmarks and tested on a third it loses to majority voting on two of three. The authors also report two unflattering findings: a generic embedding baseline wins on synthetic PrOntoQA, and cross-model answer-agreement is a strong signal by itself.

Strengths: The evaluation is built so as not to fool itself (grouped CV, a fixed classifier for the paired tests, one confound knocked out per experiment), and the paper reports its own negative results (OOD non-transfer, the PrOntoQA loss, the inert features) without hedging. The Section 4.3 story, where they noticed the forward/reverse "energy gap" features are identically zero under symmetric cosine distance and replaced them, is the kind of detail that makes me trust the rest.

Weaknesses: Everything rests on one embedding model and one sentence splitter, neither varied; the analysis is correlational; and several tables, including the 200-problem OOD splits, have no error bars.

**Audience:**

Yes

**Audience Explanation:**

Anyone working on CoT verification, selective prediction, or trace reranking will care that a cheap text-only signal recovers most of a heavier embedding classifier and beats majority voting in-distribution. Beyond the features, the paper is a clean template for measuring a trace-level correctness signal without leaking problem identity, which is a common mistake in this space. And the negative results (the verifier not transferring, engineered geometry helping on naturalistic but not synthetic reasoning) are exactly the boundary conditions a practitioner wants before deploying something like this.

**Broader Impact Concerns:**

The existing statement is good and does not need expanding.

**Claims And Evidence:**

Yes

**Claims Explanation:**

The claims hold because they are tightly scoped and then defended against the obvious confounds. Grouped CV plus label permutation handles leakage; the length ablation handles verbosity; the geometry-only subset (Table 4) handles "any interpretable feature would do." The verifier section gives the good in-distribution number and the bad OOD number equal weight, and the paper never claims mechanism: Section 7 is explicitly a sensitivity probe, not a causal result. I did not find a claim that outruns its evidence. The embedding-backbone question below affects how far the result generalizes, not whether what is written is supported.

**Requested Changes:**

None are required for my recommendation for acceptance. In rough order of how much they would raise my confidence:
1. Vary the embedding backbone (ideally the sentence splitter too). "Trajectory geometry" is defined entirely through all-MiniLM-L6-v2. Re-running on one more sentence-transformer, even on a single benchmark, and showing the gap over the baselines survives would turn the biggest caveat into evidence. This is the change I most want to see.
2. Add CIs to Tables 3, 6, 7, 8. The bootstrap is already in place. The OOD splits are 200 problems each, so the FOLIO row (OOD $0.765$ vs majority $0.740$) is hard to read without an interval, and the in-dist/OOD inversion would be far more convincing with error bars.
3. One figure of the geometric intuition. A 2D projection of a few correct vs incorrect traces (the wandering-vs-direct picture from the intro) would ground a story that currently lives mostly in feature names.

---

> ### Author Response · Authors · 2026-06-30
>
> We thank the reviewer for the helpful and detailed feedback. While preparing the revision we caught and fixed a data-integrity bug in our own pipeline: the PrOntoQA gold field was a constant string, so its precomputed correctness flag checked whether the model emitted that string rather than whether the answer was right. We replaced it with an independent oracle that recomputes gold by forward-chaining each problem's stated ontology, validated against 15 of 15 independent hand and agent re-derivations and the strong-model majority on 200 of 200 problems (Appendix D, footnote p.3). The correction raises the headline from 0.748 to 0.754 and widens every paired significance margin.
>
> Because your assessment turned on the negative results being reported straight, we flag the two effects that cut the other way. (1) It removed a finding that was an artifact, the out-of-distribution verifier collapse on PrOntoQA; corrected, the verifier and majority voting fall within a few points on all three held-out benchmarks, with GSM8K the lone clear loss (Section 6.3), and we make no transfer claim. (2) The same relabeling improved in-distribution majority voting, narrowing the verifier's in-distribution lead to 5.3 points (0.888 vs 0.835).
>
> > Vary the embedding backbone (ideally the sentence splitter too). "Trajectory geometry" is defined entirely through all-MiniLM-L6-v2. Re-running on one more sentence-transformer, even on a single benchmark, and showing the gap over the baselines survives would turn the biggest caveat into evidence. This is the change I most want to see.
>
> This was the change you most wanted to see, and it became the central one. We re-extracted every feature and re-ran the full evaluation under two further encoders from different families, all-mpnet-base-v2 and e5-base-v2 (Section 5.2.1, Table 6); the ordering of the full engineered set over the cheap baselines holds on all three. Re-running it also pushed us to test a generic order-aware model over the same sequences, which changed the paper's central claim: the geometric subset (`geometry_only`) only matches a generic embedding once the backbone is varied, and which of the two leads flips by encoder, so we retired the "specifically geometric" framing you saw at submission. In its place, a within-model ablation isolates the predictive content as specifically order (+0.081), not capacity (+0.005) or set structure (+0.028), holding the model and its inputs fixed (Section 5.2, Table 5, Figure 1); the geometric features are now a cheap, interpretable recovery of that order signal rather than the property itself. The order finding replicates on e5 (GRU 0.833). We did not vary the sentence splitter, and we name it as the remaining backbone-side limitation (Section 9).
>
> > Add CIs to Tables 3, 6, 7, 8. The bootstrap is already in place. The OOD splits are 200 problems each, so the FOLIO row (OOD 0.765 vs majority 0.740) is hard to read without an interval, and the in-dist/OOD inversion would be far more convincing with error bars.
>
> The bootstrap now resamples whole problems rather than individual traces, and the clustered 95% CIs are in Table 15 (per-benchmark and leave-one-dataset-out) and Table 16 (reranking); Table 7 already carried them. The held-out FOLIO row you single out is the near-tie that most needed an interval: after the correction it is 0.745 vs 0.740, with verifier [0.685, 0.805] and majority [0.675, 0.795] overlapping (Table 16). The in-distribution versus out-of-distribution contrast now reads from the intervals: in distribution the verifier [0.863, 0.913] sits above majority [0.807, 0.863], meeting only at the boundary, while out of distribution the two overlap on every held-out benchmark. We did not add intervals to the backbone table (Table 6): its cells report the best of three classifier families, varying by cell, while our clustered CIs are computed on a fixed random forest (Section 5.3), so an interval next to a best-of-three point would bracket a different estimator than the one shown. The backbone results are reported as point estimates; the ordering holds on all three encoders, with the e5 margin the smallest at +0.019, and we can add a fixed-random-forest backbone table during discussion if you would find it useful.

---

> > ### Author Response · Authors · 2026-06-30
> >
> > > One figure of the geometric intuition. A 2D projection of a few correct vs incorrect traces (the wandering-vs-direct picture from the intro) would ground a story that currently lives mostly in feature names.
> >
> > We tried this first: a 2D PCA of a few correct against a few incorrect traces. It was not representative of the corpus, since some incorrect traces were more compact than some correct ones, so a hand-picked panel would have shown a separation the population does not support, and projecting per-sentence positions discards the order information that is the actual finding. We replaced it with an aggregate figure over all 2,709 traces with at least three sentences (Figure 2): drift from the opening premise by normalized position, split by correctness. Incorrect traces sit measurably farther from the premise through most of the trajectory, with separating 95% bands. This is the same wandering-versus-direct intuition at population scale rather than from chosen examples.
> >
> > We are happy to provide any of these additional analyses if useful.